# Covid-19 Image Classification With Image Enhancement and Transfer Learning

## Abstract

Traditional methods and clinical practice of medical image classification have reached their limit regarding performance, making it difficult to improve through normal means. The emergence of deep neural networks such as the convolutional neural network (CNN) have proven to be an effective method on varying image classification tasks. To improve the performance of these methods, we propose to use image enhancement techniques to improve the quality and perception of medical images and to reduce inherent noise. We propose to use two spatial domain image enhancement techniques as a preprocessing step, median filter, a filtering technique, and contrast limited adaptive histogram equalization, a contrast enhancement technique, on Covid-19 CT scans before processing them through pretrained transfer learning algorithms. Through this experiment, we will demonstrate the effect of image enhancement on classifying and predicting Covid-19 images and diagnosis.

## 1 Introduction

As we see another variant of the SARS-CoV-2 virus Omicron and its big impact, the new Coronavirus disease (Covid-19) is making its undeniable influence all over the world. However, this unfortunate event has shown to be a good opportunity for research in medical imaging, notably application of deep learning methods for medical diagnosis. Medical imaging is different technologies that are used to visualize the human body with the purpose of monitoring, diagnosing or treating medical conditions [1]. A common one is Computed Tomography (CT) which is a noninvasive medical examination that uses specialized X-ray to produce cross-sectional images of the body [2]. With the pandemic, many were interested in using deep learning models to classify and diagnose Covid-19 CT scans. However, medical diagnosis depends on image acquisition where the quality of the image can get affected by different kinds of noise, hence affecting the diagnosis at the same time [3].

The recent growth of deep learning such as convolutional neural networks (CNN) is having an enormous impact on image classification/recognition modeling especially in medical imaging. However these models require a big amount of training data, and because of the need of high professional expertise to label medical images, medical image datasets are hard to collect and to create. Also, during the image acquisition phase, medical images can get affected by different noises and hence create difficulties when passed through processing steps [4].

In this study, we aim to use spatial domain image enhancement techniques, specifically median filter and contrast limited adaptive histogram equalization (CLAHE) to enhance the quality of Covid 19 CT scans as input images and to observe the impact on transfer learning models. These two methods were deemed to be better than other similar methods such as mean filter or histogram equalization [3]. Image enhancement is the process of improving digital image quality by bringing out details that are obscured or hidden. Spatial domain enhancement methods are a type of enhancement that perform

operations directly on the image pixels. In fact, spatial techniques are used to directly change the gray level values of individual pixels determined by the gray values of the points within a neighborhood around the pixel and thus altering the overall contrast of the image [5].

We make the following contributions:

- We use spatial domain image enhancement techniques to improve the quality of CT scans as a preprocessing step. This includes applying a median filter and applying a contrast limited adaptive histogram equalization on the images.

- We trained and evaluated transfer learning convolutional networks such as VGG19, ResNet50 and DenseNet121 as baseline models to demonstrate if the difference in model performance is significant.

- We demonstrate the effectiveness of our method in predicting/diagnosing Covid-19 based on open public datasets.

## 2  Background

Among filtering techniques, median filter is a non linear filtering technique where the center pixel of a NxN neighborhood is replaced by the median value of the corresponding window. By doing so, the median filter allows noise reduction. Furthermore, in contrast to other filtering techniques such as the mean filter, median filter also helps to preserve useful details for future image processing. These filters used for reducing noise are also called smoothing filters in image processing [6].

Another spatial domain enhancement technique is the adaptive histogram equalization (AHE) which is used to improve contrast in images through the histogram of the image. An image histogram is a gray-scale value distribution that shows the frequency of occurrence of each gray-level value [7]. Adaptive histogram equalization computes multiple histograms, each corresponding to different sections of the image and tries to distribute the frequency of each histogram to be more even. Adaptive histogram equalization is better than just ordinary histogram equalization because ordinary histogram equalization only considers the global contrast of the image which in many cases may not be the best since it does not preserve the image details. However, for AHE, if noise is present in small areas, it will get amplified. That is why we can apply a contrast limit that will clip any histogram bin that is above a certain contrast limit and distribute those pixels uniformly to the other bins before applying histogram equalization [8].

In computer vision, transfer learning has become an undeniable help for image recognition. The idea behind transfer learning is to use pretrained models, which were trained on a large dataset, to then transfer that knowledge. For image classification, transfer learning is used by freezing the early convolutional layers of the network and only training the last few layers. The thought behind is that the convolutional layers extract general features such as edges, patterns, gradients and the later layers extract specific features of the image. Hence, training the new network with pre-trained weights will help reduce the computational power needed to train the network and to speed up the learning process [9].

## 3  Datasets

We use two datasets for our study. The first is the Covid-CT dataset whose utility had been confirmed by a senior radiologist in Tongji Hospital, Wuhan, China, who had performed a large number of Covid-19 diagnoses. It collected 760 preprints about Covid-19 from medRxiv and bioRxiv for collection of positive Covid-19 CT images and used other databases for non-Covid CT images such as MedPix, LUNA, etc. CT scans were manually selected from the preprints and were judged to be either Covid CT scan or non-Covid CT scan. In the end, the CT scan dataset was composed of 349 CT images labeled as being positive for Covid-19 and 397 CT images labeled as being negative for Covid-19 [10].

The second dataset is the publicly available SARS-COV-2 dataset whose data have been collected directly from real patients in hospitals from Sao-Paulo, Brazil. The dataset is composed of 1252 CT images that were labeled positive for Covid-19 and 1230 CT images that were labeled as negative for Covid-19. We combined the two datasets for the training of our model with a total of 3228 CT scans [11].

# 4 Methodology

The general methodology used in our study consists of first taking input images of CT scans of shape 224 x 224 in RGB colors and applying normalization. Image enhancement (Median blur or CLAHE) will then be applied on the normalized images. When training, we used image data augmentation and transfer learning. At last, we evaluate our model's performance. To compare the baseline models, we use 3-fold cross validation in order to use statistical testing to observe if there is significant difference in the performance of our model with the proposed idea. Below is a detailed explanation of our methodology.

## 4.1 Transfer Learning

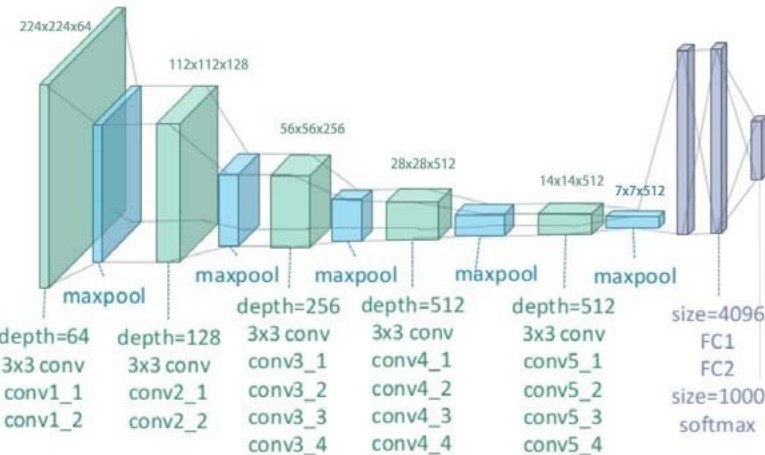

Figure 1: VGG19 architecture (Image source: reaserchgate.net)

Figure 1 is the architecture of the VGG19 model, one of the baseline models used in our study (VGG19, ResNet50, DenseNet121). To implement transfer learning, we loaded the pretrained CNN model with the weights used to train on ImageNet and we replaced the classifier with our own classifier where the last layer has only two output neurons, one for Covid and one for non-Covid. Also, we froze all the convolutional layers of our pretrained model so that we only train on the parameters of our own classifier on our own dataset.

To do so, we used a Flatten layer to convert the output of the convolutional part of our pre-trained CNN model into a 1D feature vector and used a Dropout layer to help prevent overfitting. The last layer which consists of our classifier was a Dense layer with the parameter units set to 2 for a binary classification and a softmax activation function for a probability output for the diagnosis. We then have our baseline models that will be used to compare with the proposed idea.

## 4.2 Median Filter

A digital image can be represented by a two-dimensional function f(x,y), and the x-y coordinates represent the spatial position information, also called the spatial domain. The process behind spatial filtering is to move the kernel point-by-point in the image function such that the center of the kernel corresponds with the point (x,y). The kernel also has its own predefined relationship, called template,

which will serve as base for calculating the kernel's response at each point [12]. For a kernel of size (2a+1, 2b+1), the output response can be calculated by the following function:

$$g(x, y) = \sum_{s=-a}^{a} \sum_{s=-b}^{b} w(s, t) f(x + s, y + t)$$

For median blur, the methodology is simply to replace a pixel value by the median value of its neighboring pixels. In fact, for every pixel of index (i,j) the result will be computed by sorting all the other pixels of index (i-1, j-1), (i-1, j), (i-1, j+1), (i, j-1), (i, j), (i, j+1), (i+1, j-1), (i+1, j) and (i+1, j+1) and the median value will be calculated from these values. The original pixel value will then be replaced by the newly calculated median [3].

In our study, we performed Median Blur with the help of OpenCV (Open Source Computer Vision Library) and used a kernel of size 5x5. We trained each baseline model (VGG19, ResNet50 and DenseNet121) with median blur as a preprocessing step and used a 3-fold cross validation to statistically test if there was a significant improvement or not.

### 4.3 CLAHE

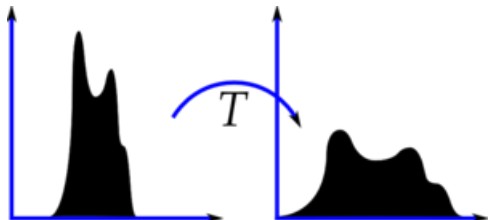

Figure 2: Histogram Equalization Process (Image source: docs.opencv.org)

Figure 3 shows 2 histograms of an image before and after equalization. In our study, we decided to use a contrast limited adaptive histogram equalization which is essentially Histogram Equalization with contrast limiting, but applied in small "tiles" instead of a global application. This will help reduce noise and keep image details better than global histogram equalization since global histogram equalization can lead to some loss of image information for example over-brightness [8].

The general histogram equalization formula is the following :

$$h(v) = round(cdf(v) - cdf_{min}) \div ((MxN) - cdf_{min}) \times (L - 1))$$

Where CDF is the cumulative distribution function, L is the maximum intensity value, M is the image width and N is the image height and h(v) is the equalized value [13].

To implement CLAHE in our study, we used OpenCV (Open Source Computer Vision Library). We first needed to have our input images have a channel of 1 where we changed the image to grayscale.After we finished applying CLAHE on our input images, we then rechanged the images channels to 3 to have RGB images for our model's training. At last,we trained our three baseline models with the enhanced images and 3-fold cross validation was also used for later statistical testing with our baseline models.

### 4.4 Data Augmentation

Image data augmentation is a simple way to artificially increase the amount of training images with random predefined transformations such as rotation, shifts, horizontal flip or many other processings at the input stage of our model's training. By doing so, our model learns how to differentiate images regardless of orientation or other transformation. This technique helps a lot for convolutional neural networks as it needs a vast amount of data.

To do so, we used the ImageDataGenerator API in Keras which generates batches of image data with real-time image augmentation.

# 5 Results

To validate our idea of the effect of image enhancement techniques and to perform predictions on Covid-19 CT scans, we performed a series of experiences. All training was done in Google Colab Pro.

## 5.1 Transfer Learning CNN Predictions

For each of our baseline models (VGG19, ResNet50, DenseNet121) and the ones with image enhancement techniques (Median Blur and CLAHE), we trained the model for 300 epochs and with batch size of 32 and ran predictions. Below are examples of different inputs of CT scans to our models:

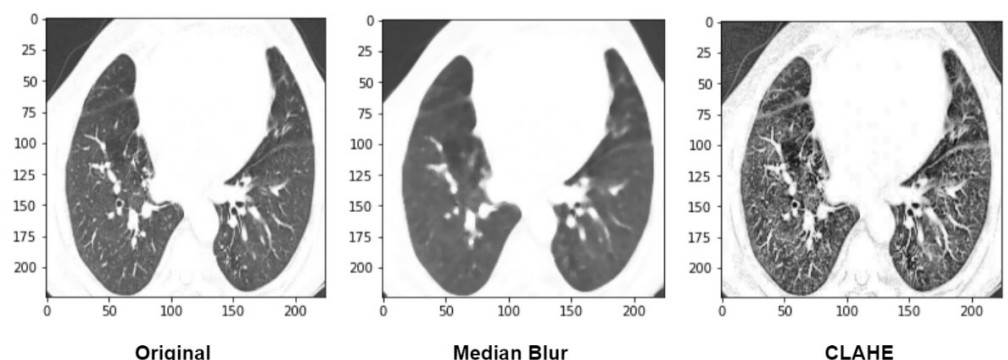

The model takes input of images of size 224 x 224. Loss is calculated using categorical cross entropy and the model uses an Adam optimizer. We summarize the performance of our different models below.

| Models | Accuracy (%) | | |
|---|---|---|---|
| | Original | Median Blur | CLAHE |
| VGG19 | 83.4 | 85.8 | 85.1 |
| ResNet50 | 77.6 | 72.4 | 78.9 |
| DenseNet121 | 89.0 | 85.1 | 87.6 |

Table 1 : Accuracy of final models

## 5.2 Models Comparison

To determine if the changes to the model's performance after applying image enhancement is statistically significant, we performed 3-fold cross-validation and trained each fold for 100 epochs. We chose only to divide our dataset into 3 fold and to use 100 epochs because of computation limits. The tables below show the results of the experiment:

| Accuracy (%) | 1 | 2 | 3 | Mean | St. dev |
|---|---|---|---|---|---|
| Original | 83.3 | 84.7 | 82.7 | 83.6 | 0.9 |
| Median Blur | 84.8 | 82.4 | 84.9 | 84.1 | 1.2 |
| CLAHE | 84.7 | 83.9 | 81.9 | 83.5 | 1.2 |

Table 2 : Accuracy in 3-fold cross-validation for VGG19

| Accuracy (%) | 1 | 2 | 3 | Mean | St. dev |
|---|---|---|---|---|---|
| Original | 71.8 | 74.5 | 74.3 | 73.5 | 1.2 |
| Median Blur | 73.8 | 71.5 | 74.8 | 73.4 | 1.4 |
| CLAHE | 50.3 | 63.8 | 65.4 | 59.8 | 6.8 |

Table 3 : Accuracy in 3-fold cross-validation for ResNet50

| Accuracy (%) | 1 | 2 | 3 | Mean | St. dev |
|---|---|---|---|---|---|
| Original | 85.7 | 85.7 | 84.5 | 85.3 | 0.5 |
| Median Blur | 77.1 | 87.7 | 82.2 | 82.4 | 4.33 |
| CLAHE | 85.2 | 83.1 | 79.4 | 82.6 | 2.4 |

Table 4 : Accuracy in 3-fold cross-validation for DenseNet121

## 6   Discussion and Future Work

With the results from the experiments using 3-fold cross-validation, we are able to perform some hypothesis testing. In fact, we use the one-sided paired T-test with $\alpha = 0.05$ to observe if a significant change was brought with the use of image enhancement. If the p-value is lower than 0.05, we reject the null hypothesis thus meaning that significant difference is present.

For VGG19, by comparing the accuracy between original images and the images enhanced by median blur and enhanced by CLAHE, we have the respective p-values : 0.38 and 0.49. Since both of the values are not smaller than 0.05, we cannot reject the null hypothesis for VGG19. In fact, even though we see a slight difference, we cannot assume that it is significant.

For ResNet50, we have the following p-values after comparing the accuracy between the original images and the ones enhanced by median blur and CLAHE : 0.46 and 0.04. For this model, since 0.46 is bigger than 0.05 we do not reject the null hypothesis for median blur, but 0.04 is smaller than 0.05, thus meaning that we do reject the null hypothesis for CLAHE. In this case, we observe that there is a significant negative change where the model with images preprocessed with CLAHE performed less well than with the original images and that no significant change is present in the performance of the model with the use of median blur.

For DenseNet121, the following p-values were calculated after performing the paired T-test on the original images and the ones enhanced by median blur and CLAHE : 0.22 and 0.09. Since both of the values are bigger than the significance level of 0.05, we conclude that neither of the enhancement techniques had a significant impact on the model's performance.

In brief, we can observe that applying image enhancement techniques does not bring significant change or improvement in the performance in comparison to our baseline models.. A possible reason and explanation behind it could be that since not all image enhancement techniques are suited for

the same task, some may be better for the task that we have chosen. Even though median blur and CLAHE did not bring improvement, there are still many image enhancement techniques that were not tested and some may have better impact. That is why it is difficult to assume that image enhancement has absolutely no impact. For future work, other image enhancement techniques can be tested, such as frequency domain enhancement techniques by using Fourier Transform. Combination of methods could also be tried as a preprocessing step or even as a data augmentation step.

# 7 Conclusion

In this paper, experiments were performed to evaluate the influence of image enhancement techniques as a preprocessing step on pre-trained models serving as baseline models and to perform diagnosis on Covid-19 CT scans. Although we have been able to perform diagnosis with our final models with an acceptable accuracy for each, our results through 3-fold cross-validation have shown that using median blur and CLAHE as an additional preprocessing step to the images did not bring significant change in regards to the accuracy of our models. In conclusion, our initial hypothesis on the impact of image enhancement technique was not verified, but future work can be done for further improvement on the models' general performance and on the analysis of the influence of image enhancement.

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
