# OpenReview forum: "Covid-19 Image Classification With Image Enhancement And Transfer Learning"
_uoft.ai/University_of_Toronto/2021/ProjectX — Submitted to ProjectX2021_

### Official Review · Reviewer_cG9H · 2022-02-09
**Established enhancement techniques used image preprocessing do not improve CT image classification with DNN**

**Rating:** 6
**Confidence:** 4

**Review:**

- The project consisted in evaluating the potential added value of preprocessing CT images (in the context of the radiological evaluation of COVID19 cases) using established image enhancement /noise reduction techniques to the performance of their classification using deep neural network systems.

- The question is potentially significant, given the tremendous interest of using DNN approaches to clinical image labelling/classification problems.

- The approach is straightforward and relatively clearly explained, with the exception of how the CT data volumes were partitioned between training, validation and testing set. I also appreciated the extraction of CT images from preprint publication but wondered whether their lesser resolution and definition, with respect to original clinical scans, would be detrimental to the enterprise.

- The results are honestly presented, showing clearly there was no beneficial effect of image preprocessing to DNN classification performances.

- Overall, I am not sure of the outcome could be expected to be different: indeed, this question must have been looked at in detail in other subfields of DNN radiology, because of the popularity gained by DNN approaches in computational radiology over the recent years, or in other fields of life sciences. A deeper search of the current literature would have been beneficial to the framing and significance of the project.

---

### Official Review · Reviewer_xpzE · 2022-02-10
**The effect of image enhancement on CNN-based COVID-19 classification on CT images**

**Rating:** 4
**Confidence:** 4

**Review:**

Summary
- This paper applies known image enhancement methods (median filter & contrast limited adaptive histogram equalization (CLAHE)) on medical CT images trying to improve classification performance for COVID-19 diagnosis. The authors use well-known CNN architectures (VGG19, ResNet50, DenseNet121) pretrained on ImageNet for classification after fine-tuning on COVID-19 data. The results show image enhancement techniques made no significant improvement over the raw images, while some types of enhancement seem to degrade the performance.

Evaluation based on ProjectX rubric
- Connection to Current Science: 1 (Shows some knowledge of the field)
- Clarity of Communication: 1 (Writing is moderately clear to follow; No visualization technique other than a simple table was used to present results; Insufficient explanation/motivation about why the particular methods were chosen, e.g. median blur & CLAHE)
- Methodological Quality: 1.5 (Applied known methods in image classification & in image enhancement to a known problem; Methods were reasonably appropriate for the problem, but some details are questionable, e.g. data augmentation; estimating variance with 3-fold split is not enough)
- Reproducibility: 0 (No open source code; Some parts of the methods are not fully specified, e.g. data augmentation)
- Total: 3.5

Major comments
- Some parts of the methods are not fully specified. Here are some examples: 1) In "4 Methodology" the authors said they applied "normalization" but did not specify what exactly is the normalization step. 2) In "4.4 Data Augmentation" there is no description of exactly what types of data augmentation were applied. Some of them would not be appropriate for CT images (see next).
- Regarding "4.4 Data Augmentaion", rotations and horizontal flips are not appropriate data augmentation for medical images, since human anatomy is not symmetric under rotation or horizontal flips (e.g. heart is on the left side).
- In "4.3 CLAHE" the authors wrote they converted the RGB images to grayscale and then changed them back to RGB colors. Clearly it's generally not possible (without any special technique) to change grayscale images to color images, so more explanation is needed about how this was done.
- In "5.2 Model Comparison", I understand the computational constraints but 3 sets are definitely not enough to estimate the variance/standard deviation. The p-values based on this small number of trials are not that meaningful.
- Most paragraphs about background materials seem to be copied from the references with minimal paraphrasing.
- Having open-sourced code accompanying the paper would be useful.

Minor comments
- This paper cites many blog articles and webpages. It'd be better to put more effort on finding the original source of the information, at least for the main  concepts used in the paper. For example, CLAHE dates back to at least 1987 [1].
- Need a better explanation/motivation about why the authors chose two image enhancement methods (median filter and CLAHE) among others.
- In "4.1 Transfer Learning" we don't need 2 neurons for COVID or non-COVID. One neuron with sigmoid activation is enough for binary classification where the output would represent the probability of having COVID (and the complement of that probability is for non-COVID).
- In "4.2 Median Filter" the authors gave a definition of spatial filters (that seemed to be copied from the webpage referenced in the paper), but this definition was actually not used for the median filter. The median cannot be represented by the equation the authors wrote.

[1] Pizer et al 1987, Adaptive histogram equalization and its variations, https://doi.org/10.1016/S0734-189X(87)80186-X

---

### Official Review · Reviewer_KMCU · 2022-02-11
**Good concept poor clinical context**

**Rating:** 5
**Confidence:** 3

**Review:**

Connection to Current Science (1.5/3)
 - Radiography is an incredibly common area for application of AI in a clinical context
 - COVID-19 is a new area of research however, its clinical diagnosis rarely includes CT
 - There is not a clear connection between this technique and how it will be used clinically - is it to be used for COVID diagnosis (if so this is not done with CT but rather a much cheaper PCR test), it is to diagnose and track COVID pneumonia (in this case their model of simply seeing if the patient has COVID does not add to the clinical picture as a diagnosis will already be known, in this situation severity of COVID pneumonia/ARDS would be needed as an outcome in their model)
 - Evidently a lack of research into COVID management and diagnosis at a medical level

Clarity of Communication (1.5/2)
 - Generally well written, occasional grammatical errors
 - Insufficient explanation in results and discussion

Methodological Quality (1.75/4)
 - The methodology of enhancing the CT and then running it through the existing radiograph searching tools is interesting however, I feel it is not well applied to a problem
 - This is potentially more useful for something where it is either hard for a physician or an existing model to identify a clinical diagnosis using current methods (e.g. some cancer presentations). Typically a patient's clinical presentation plus x-ray findings are enough to monitor COVID induced ARDS as this is generally very stark finding on radiographs.
 - Even if there is a subclinical case of ARDS caused by COVID-19 making this diagnosis from radiographs would be unlikely to change clinical management
 - Methods were generally described well and the overall structure was not too complex

Reproducibility (0.75/1)
 - Used open source databases and as well as existing enhancement tools
 - Potentially less work than would be expected over a 5 month time frame

---

### Official Review · Reviewer_sqUH · 2022-02-14
**Review of "Covid-19 Image Classification With Image Enhancement and Transfer Learning"**

**Rating:** 6
**Confidence:** 3

**Review:**

1) Connection to Current Science (science and practice) [SCALE 0(low) – 3(high)]

•	Does this work show knowledge of the existing state of the field? – 0.5
•	Does this work add something new to the literature? - 1
•	Do the teams discuss what their pathway to implementation looks like? - 0
Comment: The idea implemented by this paper is fantastic. However, I don’t have a great understanding of the severity of the issue they are discussing: “However, medical diagnosis depends on image acquisition where the quality of the image can get affected by different kinds of noise, hence affecting the diagnosis at the same time”. What are the impacts of this? How dire is this problem? Is it resulting in a lack of diagnosis? False diagnosis? What would the impact on patient health be if we could improve the automated image diagnosis of COVID? More information on how this would be translated into the clinical setting if improvement in classification would also have been helpful. I just can’t assess the impact/novelty of this project with the information given to me.
Final score – 1.5

2) Clarity of Communication [SCALE 0(low) – 2(high)]
•	Is the writing clear and easy to follow? – yes
•	Are data visualization techniques chosen and labelled well? – I would have chosen a graphical way instead of tables to compare accuracy (tables 1-4)
•	Is there a clear logical structure to the paper? – Yes
Comment: Why was model accuracy chosen from the plethora of machine learning metrics available?
Final score – 1.5

3) Methodological Quality [SCALE 0 (overly complicated models that are ill-suited to the problem) - 4 (develops new methods that provide insight into an important problem)]
•	Is this paper making reasonable assumptions? yes
•	Does this paper use methods that are appropriate to the problem at hand? yes
•	Does the paper introduce new and interesting methods? (old methods applied well is good as well) yes
•	Does the paper avoid needless model complexity? Yes
Comment: This paper has strong methods and applications. It is the greatest strength overall of the paper
Final Score – 3

4) Reproducibility [SCALE 0(low) to 1(open source code)]
•	Does this paper seem reasonable as work conducted in a span of 5 months? yes
•	Has the team been open for others to reproduce the paper’s results? No open-source code is available.
Final Score – 0.5

---

### Official Review · Reviewer_8qgc · 2022-02-15
**Review—Covid-19 Image Classification With Image Enhancement And Transfer Learning**

**Rating:** 7
**Confidence:** 3

**Review:**

**Connection to Current Science (1.5/3)**

- Demonstrates knowledge of the field and builds on previous research
- Study area is timely for the COVID-19 pandemic

**Clarity of Communication (1.75/2)**

- Authors describe rationale, methods, and results clearly
- Methods are described with sufficient detail for a non-technical audience
- Figures and tables are clear and provide succinct summary of concepts of and results

**Methodological Quality (3.25/4)**

- Builds on previous work to adapt pre-trained models for COVID-19 classification problem
- Used transfer learning to adapt previous work and image enhancement techniques to reduce image noise
- Given the health context, other measures of predictive performance would have been valuable (e.g., sensitive and specificity)—false negatives can jeopardize patient treatment and false positives can have negative impacts on an already strained health system
- Suggest evaluating and mentioning the data quality from the Chinese and Brazilian CT cans. The CT scans were labelled as positive and negative, but was the original labelling validated against a diagnostic gold standard? Measurement error from original data may carry over bias into the final prediction model.
- Prediction intervals would help access prediction precision
- Consider impact of data shift as COVID-19 variants can present different symptoms (i.e., CT cans from early in the pandemic may not look like new CT scans with newer variants that are less efficient at infecting lungs)

**Reproducibility (0.25/1)**

- Used publicly available data
- No supplementary materials provided to assess reproducibility

---

### Decision · Program_Chairs · 2022-02-19

NA